# Hierarchical Methods of Moments

**Matteo Ruffini** *
Universitat Politècnica
de Catalunya

**Guillaume Rabusseau** †
McGill University

**Borja Balle** ‡
Amazon Research
Cambridge

## Abstract

Spectral methods of moments provide a powerful tool for learning the parameters of latent variable models. Despite their theoretical appeal, the applicability of these methods to real data is still limited due to a lack of robustness to model misspecification. In this paper we present a hierarchical approach to methods of moments to circumvent such limitations. Our method is based on replacing the tensor decomposition step used in previous algorithms with approximate joint diagonalization. Experiments on topic modeling show that our method outperforms previous tensor decomposition methods in terms of speed and model quality.

## 1 Introduction

Unsupervised learning of latent variable models is a fundamental machine learning problem. Algorithms for learning a variety of latent variable models, including topic models, hidden Markov models, and mixture models are routinely used in practical applications for solving tasks ranging from representation learning to exploratory data analysis. For practitioners faced with the problem of training a latent variable model, the decades-old Expectation-Maximization (EM) algorithm [1] is still the tool of choice. Despite its theoretical limitations, EM owes its appeal to (i) the robustness of the maximum-likelihood principle to model misspecification, and (ii) the need, in most cases, to tune a single parameter: the dimension of the latent variables.

On the other hand, method of moments (MoM) algorithms for learning latent variable models via efficient tensor factorization algorithms have been proposed in the last few years [2–9]. Compared to EM, moment-based algorithms provide a stronger theoretical foundation for learning latent variable models. In particular, it is known that in the realizable setting the output of a MoM algorithm will converge to the parameters of the true model as the amount of training data increases. Furthermore, MoM algorithms only make a single pass over the training data, are highly parallelizable, and always terminate in polynomial time. However, despite their apparent advantages over EM, the adoption of MoM algorithms in practical applications is still limited.

Empirical studies indicate that initializing EM with the output of a MoM algorithm can improve the convergence speed of EM by several orders of magnitude, yielding a very efficient strategy to accurately learn latent variable models [8–10]. In the case of relatively simple models this approach can be backed by intricate theoretical analyses [11]. Nonetheless, these strategies are not widely deployed in practice either.

The main reason why MoM algorithms are not adopted by practitioners is their lack of robustness to model misspecification. Even when combined with EM, MoM algorithms fail to provide an initial estimate for the parameters of a model leading to fast convergence when the learning problem is too far from the realizable setting. For example, this happens when the number of the latent variables used in a MoM algorithm is too small to accurately represent the training data. In contrast, the model

obtained by standalone EM in this case is reasonable and desirable: when asked for a small number of latent variables EM yields a model which is easy to interpret and can be useful for data visualization and exploration. For example, an important application of low-dimensional learning can be found in mixture models, where latent class assignments provided by a simple model can be used to split the training data into disjoint datasets to which EM is applied recursively to produce a hierarchical clustering [12, 13]. The tree produced by such clusterings procedure provides a useful aid in data exploration and visualization even if the models learned at each branching point do not accurately represent the training data.

In this paper we develop a hierarchical method of moments that produces meaningful results even in misspecified settings. Our approach is different from previous attemps to design MoM algorithms for misspecified models. Instead of looking for convex relaxations of existing MoM algorithms like in [14–16] or analyzing the behavior of a MoM algorithm with a misspecified number of latent states like in [17, 18], we generalize well-known simultaneous diagonalization approaches to tensor decomposition by phrasing the problem as a non-convex optimization problem. Despite its non-convexity, the hierarchical nature of our method allows for a fast accurate solution based on low-dimensional grid search. We test our method on synthetic and real-world datasets on the topic modeling task, showcasing the advantages of our approach and obtaining meaningful results.

## 2 Moments, Tensors, and Latent Variable Models

This section starts by recalling the basic ideas behind methods of moments for learning latent variable models via tensor decompositions. Then we review existing tensor decomposition algorithms and discuss the effect of model misspecification on the output of such algorithms.

For simplicity we consider first a *single topic model* with $k$ topics over a vocabulary with $d$ words. A single topic model defines a generative process for text documents where first a topic $Y \in [k]$ is drawn from some discrete distribution $\mathbb{P}[Y = i] = \omega_i$, and then each word $X_t \in [d]$, $1 \le t \le T$, in a document of length $T$ is independently drawn from some distribution $\mathbb{P}[X_t = j | Y = i] = \mu_{i,j}$ over words conditioned on the document topic. The model is completely specified by the vector of topic proportions $\omega \in \mathbb{R}^k$ and the word distributions $\mu_i \in \mathbb{R}^d$ for each topic $i \in [k]$. We collect the word distributions of the model as the columns of a matrix $M = [\mu_1 \cdots \mu_k] \in \mathbb{R}^{d \times k}$.

It is convenient to represent the words in a document using one-hot encodings so that $X_t \in \mathbb{R}^d$ is an indicator vector. With this notation, the conditional expectation of any word in a document drawn from topic $i$ is $\mathbb{E}[X_t | Y = i] = \mu_i$, and the random vector $X = \sum_{t=1}^T X_t$ is conditionally distributed as a multinomial random variable, with parameters $\mu_i$ and $T$. Integrating over topics drawn from $\omega$ we obtain the first moment of the distribution over words $M_1 = \mathbb{E}[X_t] = \sum_i \omega_i \mu_i = M\omega$. Generalizing this argument to pairs and triples of distinct words in a document yields the matrix of second order moments and the tensor of third order moments of a single topic model:

$$M_2 = \mathbb{E}[X_s \otimes X_t] = \sum_i \omega_i \mu_i \otimes \mu_i \in \mathbb{R}^{d \times d} \ , \tag{1}$$

$$M_3 = \mathbb{E}[X_r \otimes X_s \otimes X_t] = \sum_i \omega_i \mu_i \otimes \mu_i \otimes \mu_i \in \mathbb{R}^{d \times d \times d} \ , \tag{2}$$

where $\otimes$ denotes the tensor (Kronecker) product between vectors. By defining the matrix $\Omega = \mathrm{diag}(\omega)$ one also obtains the expression $M_2 = M\Omega M^\top$.

A method of moments for learning single topic models proceeds by (i) using a collection of $n$ documents to compute empirical estimates $\hat{M}_1$, $\hat{M}_2$, $\hat{M}_3$ of the moments, and (ii) using matrix and tensor decomposition methods to (approximately) factor these empirical moments and extract the model parameters from their decompositions. From the algorithmic point of view, the appeal of this scheme resides in the fact that step (i) requires a single pass over the data which can be trivially parallelized using map-reduce primitives, while step (ii) only requires linear algebra operations whose running time is independent of $n$. The specifics of step (ii) will be discussed in Section 2.1.

Estimating moments $\hat{M}_m$ from data with the property that $\mathbb{E}[\hat{M}_m] = M_m$ for $m \in \{1, 2, 3\}$ is the essential requirement for step (i). In the case of single topic models, and more generally multi-view models, such estimations are straightforward. For example, a simple consistent estimator takes a collection of documents $\{x^{(i)}\}_{i=1}^n$ and computes $\hat{M}_3 = (1/n) \sum_{i=1}^n x_1^{(i)} \otimes x_2^{(i)} \otimes x_3^{(i)}$ using the

first three words from each document. More data-efficient estimators for datasets containing long documents can be found in the literature [19].

For more complex models the method sketched above requires some modifications. Specifically, it is often necessary to correct the statistics directly observable from data in order to obtain vectors/matrices/tensors whose expectation over a training dataset exhibits precisely the relation with the parameters $\omega$ and $M$ described above. For example, this is the case for Latent Dirichlet Allocation and mixtures of spherical Gaussians [4, 6]. For models with temporal dependence between observations, e.g. hidden Markov models, the method requires a spectral projection of observables to obtain moments behaving in a multi-view-like fashion [3, 20]. Nonetheless, methods of moments for these models and many others always reduces to the factorization of a matrix and tensor of the form $M_2$ and $M_3$ given above.

## 2.1 Existing Tensor Decomposition Algorithms

Mathematically speaking, methods of moments attempt to solve the polynomial equations in $\omega$ and $M$ arising from plugging the empirical estimates $\hat{M}_m$ into the expressions for their expectations given above. Several approaches have been proposed to solve these non-linear systems of equations.

A popular method for tensor decomposition is Alternating Least Squares (ALS) [21]. Starting from a random initialization of the factors composing a tensor, ALS iteratively fixes two of the three factors, and updates the remaining one by solving an overdetermined linear least squares problem. ALS is easy to implement and to understand, but is known to be prone to local minima, needing several random restarts to yield meaningful results. These limitations fostered the research for methods with guarantees, that, in the unperturbed setting, optimally decompose a tensor like the one in Eq. (2). We now briefly analyze some of these methods.

The tensor power method (TPM) [2] starts with a whitening step where, given the SVD $M_2 = USU^\top$, the whitening matrix $E = US^{1/2} \in \mathbb{R}^{d \times k}$ is used to transform $M_3$ into a symmetric orthogonally decomposable tensor

$$T = \sum_{i=1}^{k} \omega_i E^\dagger \mu_i \otimes E^\dagger \mu_i \otimes E^\dagger \mu_i \in \mathbb{R}^{k \times k \times k} \tag{3}$$

The weights $\omega_i$ and vectors $\mu_i$ are then recovered from $T$ using a tensor power method and inverting the whitening step.

The same whitening matrix is used in [3, 4], where the authors observe that the whitened slices of $M_3$ are simultaneously diagonalized by the Moore-Penrose pseudoinverse of $M\Omega^{1/2}$. Indeed, since $M_2 = M\Omega M^\top = EE^\top$, there exists a unique orthonormal matrix $O \in \mathbb{R}^{k \times k}$ such that $M\Omega^{1/2} = EO$. Writing $M_{3,r} \in \mathbb{R}^{d \times d}$ for the $r$th slice of $M_3$ across its second mode and $m_r$ for the $r$th row of $M$, it follows that

$$M_{3,r} = M\Omega^{1/2} diag(m_r) \Omega^{1/2} M^\top = EO diag(m_r) O^\top E^\top \ .$$

Thus, the problem can be reduced to searching for the common diagonalizer $O$ of the whitened slices of $M_3$ defined as

$$H_r = E^\dagger M_{3,r} E^{\dagger\top} = O diag(m_r) O^\top \ . \tag{4}$$

In the noiseless settings it is sufficient to diagonalize any of the slices $M_{3,r}$. However, one can also recover $O$ as the eigenvectors of a random linear combination of the various $H_r$ which is more robust to noise [3].

Lastly, the method proposed in [22] consists in directly performing simultaneous diagonalization of random linear combinations of slices of $M_3$ without any whitening step. This method, which in practice is slower than the others (see Section 4.1), under an incoherence assumption on the vectors $\mu_i$, can robustly recover the weights $\omega_i$ and vectors $\mu_i$ from the tensor $M_3$, even when it is not orthogonally decomposable.

## 2.2 The Misspecified Setting

The methods listed in the previous section have been analyzed in the case where the algorithm only has access to noisy estimates of the moments. However, such analyses assume that the data was

generated by a model from the hypothesis class, that the matrix $M$ has rank $k$, and that this rank is known to the algorithm. In practice the dimension $k$ of the latent variable can be cross-validated, but in many cases this is not enough: data may come from a model outside the class, or from a model with a very large true $k$. Besides, the moment estimates might be too noisy to provide reliabe estimates for large number of latent variables. It is thus frequent to use these algorithms to estimate $l < k$ latent variables. However, existing algorithms are not robust in this setting, as they have not been designed to work in this regime, and there is no theoretical explanation of what their outputs will be.

The methods relying on a whitening step [2–4], will perform the whitening using the matrix $E_l^\dagger$ obtained from the low-rank SVD truncated at rank $l$: $M_2 \approx U_l S_l U_l^\top = E_l E_l^\top$. TPM will use $E_l$ to whiten the tensor $M_3$ to a tensor $T_l \in \mathbb{R}^{l \times l \times l}$. However, when $k > l$, $T_l$ may not admit a symmetric orthogonal decomposition [4]. Consequently, it is not clear what TPM will return in this case and there are no guarantees it will even converge. The methods from [3, 4] will compute the matrices $H_{l,r} = E_l^\dagger M_{3,r} E_l^{\dagger\top}$ for $r \in [d]$ that may not be jointly diagonalizable, and in this case there is no theoretical justification of what what the result of these algorithms will be. Similarly, the simultaneous diagonalization method proposed in [22] produces a matrix that nearly diagonalizes the slices of $M_3$, but no analysis is given for this setting.

## 3 Simultaneous Diagonalization Based on Whitening and Optimization

This section presents the main contribution of the paper: a simultaneous diagonalization algorithms based on whitening and optimization we call SIDIWO (**Si**multaneous **Di**agonalization based on **W**hitening and **O**ptimization). When asked to produce $l = k$ components in the noiseless setting, SIDIWO will return the same output as any of the methods discussed in Section 2.1. However, in contrast with those methods, SIDIWO will provide useful results with a clear interpretation even in a misspecified setting ($l < k$).

### 3.1 SIDIWO in the Realizable Setting

To derive our SIDIWO algorithm we first observe that in the noiseless setting and when $l = k$, the pair $(M, \omega)$ returned by all methods described in Section 2.1 is the solution of the optimization problem given in the following lemma[5].

**Lemma 3.1** *Let $M_{3,r}$ be the $r$-th slice across the second mode of the tensor $M_3$ from (2) with parameters $(M, \omega)$. Suppose $rank(M) = k$ and let $\Omega = \mathrm{diag}(\omega)$. Then the matrix $(M\Omega^{1/2})^\dagger$ is the unique optimum (up to column rescaling) of the optimization problem*

$$\min_{D \in \mathcal{D}_k} \sum_{i \neq j} \left( \sum_{r=1}^{d} (DM_{3,r}D^\top)_{i,j}^2 \right)^{1/2} , \tag{5}$$

*where $\mathcal{D}_k = \{D : D = (EO_k)^\dagger \text{ for some } O_k \text{ s.t. } O_k O_k^\top = \mathbb{I}_k\}$ and $E$ is the whitening matrix defined in Section 2.1.*

**Remark 1 (The role of the constraint)** *Consider the cost function of Problem (5): in an unconstrained setting, there may be several matrices minimizing that cost. A trivial example is the zero matrix. A less trivial example is when the rows of $D$ belong to the orthogonal complement of the column space of the matrix $M$. The constraint $D = (EO_k)^\dagger$ for some orthonormal matrix $O_k$ first excludes the zero matrix from the set of feasible solutions, and second guarantees that all feasible solutions lay in the space generated by the columns of $M$.*

Problem (5) opens a new perspective on using simultaneous diagonalization to learn the parameters of a latent variable model. In fact, one could recover the pair $(M, \omega)$ from the relation $M\Omega^{1/2} = D^\dagger$ by first finding the optimal $D$ and then individually retrieving $M$ and $\omega$ by solving a linear system using the vector $M_1$. This approach, outlined in Algorithm 1, is an alternative to the ones presented in the literature up to now (even though in the noiseless, realizable setting, it will provide the same

**Algorithm 1** SIDIWO: Simultaneous Diagonalization based on Whitening and Optimization

---

**Require:** $M_1$, $M_2$, $M_3$, the number of latent states $l$
 1: Compute a SVD of $M_2$ truncated at the $l$-th singular vector: $M_2 \approx U_l S_l U_l^\top$.
 2: Define the matrix $E_l = U_l S_l^{1/2} \in \mathbb{R}^{d \times l}$.
 3: Find the matrix $D \in \mathcal{D}_l$ optimizing Problem (5).
 4: Find $(\tilde{M}, \tilde{\omega})$ solving $\begin{cases} \tilde{M}\tilde{\Omega}^{1/2} = D^\dagger \\ \tilde{M}\tilde{\omega}^\top = M_1 \end{cases}$
 5: **return** $(\tilde{M}, \tilde{\omega})$

---

results). Similarly to existing methods, this approach requires to know the number of latent states. We will however see in the next section that Algorithm 1 provides meaningful results even when a misspecified number of latent states $l < k$ is used.

## 3.2 The Misspecified Setting

Algorithm 1 requires as inputs the low order moments $M_1$, $M_2$, $M_3$ along with the desired number of latent states $l$ to recover. If $l = k$, it will return the exact model parameters $(M, \omega)$; we will now see that it will also provide meaningful results when $l < k$. In this setting, Algorithm 1 returns a pair $(\tilde{M}, \tilde{\omega}) \in \mathbb{R}^{d \times l} \times \mathbb{R}^l$ such that the matrix $D = (\tilde{M}\tilde{\Omega}^{1/2})^\dagger$ is optimal for the optimization problem

$$\min_{D \in \mathcal{D}_l} \sum_{i \neq j} \left( \sum_{r=1}^d (DM_{3,r}D^\top)_{i,j}^2 \right)^{1/2}. \tag{6}$$

Analyzing the space of feasible solutions (Theorem 3.1) and the optimization function (Theorem 3.2), we will obtain theoretical guarantees on what SIDIWO returns when $l < k$, showing that the trivial solutions are not feasible, and that, in the space of feasible solutions, SIDIWO's optima will approximate the true model parameters according to an intuitive geometric interpretation.

**Remarks on the constraints.** The first step consists in analyzing the space of feasible solutions $\mathcal{D}_l$ when $l < k$. The observations outlined in Remark 1 still hold in this setting: the zero solution and the matrices laying in the orthonormal complement of $M$ are not feasible. Furthermore, the following theorem shows that other undesirable solutions will be avoided.

**Theorem 3.1** *Let $D \in \mathcal{D}_l$ with rows $d_1, ..., d_l$, and let $\mathbb{I}_{r,s}$ denote the $r \times s$ identity matrix. The following facts hold under the hypotheses of Lemma 3.1:*

 1. *For any row $d_i$, there exists at least one column of $M$ such that $\langle d_i, \mu_j \rangle \neq 0$.*

 2. *The columns of any $\tilde{M}$ satisfying $\tilde{M}\tilde{\Omega}^{1/2} = D^\dagger$ are a linear combination of those of $M$, laying in the best-fit $l$-dimensional subspace of the space spanned by the columns of $M$.*

 3. *Let $\pi$ be any permutation of $\{1, ..., d\}$, and let $M_\pi$ and $\Omega_\pi$ be obtained by permuting the columns of $M$ and $\Omega$ according to $\pi$. If $\langle \mu_i, \mu_j \rangle \neq 0$ for any $i, j$, then $((M_\pi\Omega_\pi^{1/2})\mathbb{I}_{k,l})^\dagger \notin \mathcal{D}_l$, and similarly $\mathbb{I}_{l,k}(M_\pi\Omega_\pi^{1/2})^\dagger \notin \mathcal{D}_l$.*

The second point of Theorem 3.1 states that the feasible solutions will lay in the best $l$-dimensional subspace approximating the one spanned by the columns of $M$. This has two interesting consequences: if the columns of $M$ are not orthogonal, point 3 guarantees that $\tilde{M}$ cannot simply be a sub-block of the original $M$, but rather a non-trivial linear combination of its columns laying in the best $l$-dimensional subspace approximating its column space. In the single topic model case with $k$ topics, when asked to recover $l < k$ topics, Algorithm 1 will not return a subset of the original $k$ topics, but a matrix $\tilde{M}$ whose columns gather the original topics via a non trivial linear combination: the original topics will all be represented in the columns of $\tilde{M}$ with different weights. When the columns of $M$ are orthogonal, this space coincides with the space of the $l$ columns of $M$ associated with the $l$ largest $\omega_i$; in this setting, the matrix $(M_\pi\Omega_\pi^{1/2})\mathbb{I}_{k,l}$ (for some permutation $\pi$) is a feasible solution and minimizes Problem (6). Thus, Algorithm 1 will recover the top $l$ topics.

**Interpreting the optima.** Let $\tilde{M}$ be such that $D = (\tilde{M}\tilde{\Omega}^{1/2})^\dagger \in \mathcal{D}_l$ is a minimizer of Problem (6). In order to better understand the relation between $\tilde{M}$ and the original matrix $M$, we will show that the cost function of Problem (6) can be written in an equivalent form, that unveils a geometric interpretation.

**Theorem 3.2** *Let $d_1, ..., d_l$ denote the rows of $D \in \mathcal{D}_l$ and introduce the following optimization problem*

$$\min_{D \in \mathcal{D}_l} \sum_{i \neq j} \sup_{v \in \mathcal{V}_M} \sum_{h=1}^{k} \langle d_i, \mu_h \rangle \langle d_j, \mu_h \rangle \omega_h v_h \qquad (7)$$

*where $\mathcal{V}_M = \{v \in \mathbb{R}^k : v = \alpha^\top M, \text{where } \|\alpha\|_2 \leq 1\}$. Then this problem is equivalent to (6).*

First, observe that the cost function in Equation (7) prefers $D$'s such that the vectors $u_i = [\langle d_i, \mu_1 \sqrt{\omega_1} \rangle, ..., \langle d_i, \mu_k \sqrt{\omega_k} \rangle]$, $i \in [l]$, have disjoint support. This is a consequence of the $\sup_{v \in \mathcal{V}_M}$, and requires that, for each $j$, the entries $\langle d_i, \mu_j \sqrt{\omega_j} \rangle$ are close zero for at least all but one of the various $d_i$. Consequently, each center will be almost orthogonal to all but one row of the optimal $D$; however the number of centers is greater than the number of rows of $D$, so the same row $d_i$ may be nonorthogonal to various centers.

For illustration, consider the single topic model: a solution $D$ to Problem (7) would have rows that should be as orthogonal as possible to some topics and as aligned as possible to the others; in other words, for a given topic $j$, the optimization problem is trying to set $\langle d_i, \mu_j \sqrt{\omega_j} \rangle = 0$ for all but one of the various $d_i$. Consequently, each column of the output $\tilde{M}$ of Algorithm 1 should be in essence aligned with some of the topics and orthogonal to the others.

It is worth mentioning that the constraint set $\mathcal{D}_l$ forbids the trivial solutions such as the zero matrix, the pseudo-inverse of any subset of $l$ columns of $M\Omega^{1/2}$, and any subset of $l$ rows of $(M\Omega^{1/2})^\dagger$ (which all have an objective value of 0).

We remark that Theorem 3.2 doesn't require the matrix $M$ to be full rank $k$: we only need it to have at least rank greater or equal to $l$, in order to guarantee that the constraint set $\mathcal{D}_l$ is well defined.

**An optimal solution when $l = 2$.** While Problem (5) can be solved in general using an extension of the Jacobi technique [23, 24], we provide a simple and efficient method for the case $l = 2$. This method will then be used to perform hierarchical topic modeling in Section 4. When $l = 2$, Equation (6) can be solved optimally with few simple steps; in fact, the following theorem shows that solving (6) is equivalent to minimizing a continuous function on the compact one-dimensional set $I = [-1, 1]$, which can easily be done by griding $I$. Using this in Step 3 of Algorithm 1, one can efficiently compute an arbitrarily good approximation of the optimal matrix $D \in \mathcal{D}_2$.

**Theorem 3.3** *Consider the continuous function $F(x) = c_1 x^4 + c_2 x^3 \sqrt{1 - x^2} + c_3 x \sqrt{1 - x^2} + c_4 x^2 + c_5$, where the coefficients $c_1, ..., c_5$ are functions of the entries of $M_2$ and $M_3$. Let $a$ be the minimizer of $F$ on $[-1, 1]$, and consider the matrix*

$$O_a = \begin{bmatrix} \sqrt{1 - a^2} & a \\ -a & \sqrt{1 - a^2} \end{bmatrix} \quad .$$

*Then, the matrix $D = (E_2 O_a)^\dagger$ is a minimizer of Problem (6) when $l = 2$.*

## 4 Case Study: Hierarchical Topic Modeling

In this section, we show how SIDIWO can be used to efficiently recover hierarchical representations of latent variable models. Given a latent variable model with $k$ states, our method allows to recover a pair $(\tilde{M}, \tilde{\omega})$ from estimate of the moments $M_1$, $M_2$ and $M_3$, where the $l$ columns of $\tilde{M}$ offer a synthetic representation of the $k$ original centers. We will refer to these $l$ vectors as *pseudo-centers*: each pseudo-center is representative of a group of the original centers. Consider the case $l = 2$. A dataset $\mathcal{C}$ of $n$ samples can be split into two smaller subsets according to their similarity to the two pseudo-centers. Formally, this assignment is done using Maximum A Posteriori (MAP) to find the pseudo-center giving maximum conditional likelihood to each sample. The splitting procedure can

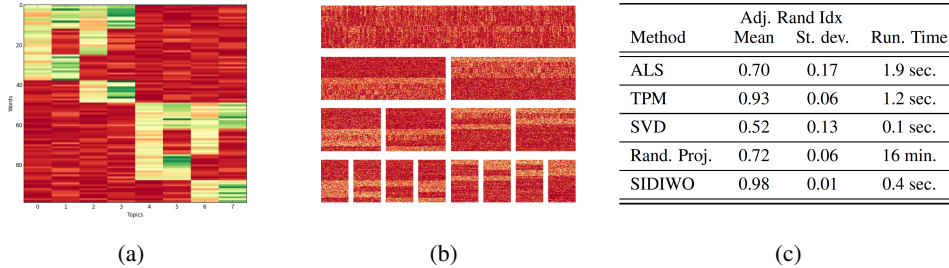

|              | Adj. Rand Idx |          |           |
| Method       | Mean | St. dev. | Run. Time |
| ------------ | ---- | -------- | --------- |
| ALS          | 0.70 | 0.17     | 1.9 sec.  |
| TPM          | 0.93 | 0.06     | 1.2 sec.  |
| SVD          | 0.52 | 0.13     | 0.1 sec.  |
| Rand. Proj.  | 0.72 | 0.06     | 16 min.   |
| SIDIWO       | 0.98 | 0.01     | 0.4 sec.  |

(a)                                (b)                                (c)

Figure 1: Figure 1a provides a visualization of the topics used to generate the sample. Figure 1b represents the hierarchy recovered with the proposed method. Table 1c reports the average and standard deviation over 10 runs of the clustering accuracy for the various methods, along with average running times.

be iterated recursively to obtain a divisive binary tree, leading to a hierarchical clustering algorithm. While this hierarchical clustering method can be applied to any latent variable model that can be learned with the tensor method of moments (e.g. Latent Dirichlet Allocation), we present it here for the single topic model for the sake of simplicity.

We consider a corpus $\mathcal{C}$ of $n$ texts encoded as in Section 2 and we split $\mathcal{C}$ into two smaller corpora according to their similarity to the two pseudo-centers in two steps: project the pseudo-centers on the simplex to obtain discrete probability distributions (using for example the method described in [25]), and use MAP assignment to assign each text $x$ to a pseudo-center. This process is summarized in Algorithm 2. Once the corpus $\mathcal{C}$ has been split into two subsets $\mathcal{C}_1$ and $\mathcal{C}_2$, each of these subsets may

---

**Algorithm 2** Splitting a corpus into two parts

---

**Require:** A corpus of texts $\mathcal{C} = (x^{(1)}, ..., x^{(n)})$.
1: Estimate $M_1$, $M_2$ and $M_3$.
2: Recover $l = 2$ pseudo-center with Algorithm 1 .
3: Project the Pseudo-center to the simplex
4: **for** $i \in [n]$ **do**
5:     Assign the text $x^{(i)}$ to the cluster $Cluster(i) = \arg\max_j \ \mathbb{P}[X = x^{(i)} | Y = j, \tilde{\omega}, \tilde{M}]$, where $\mathbb{P}[X | Y = j, \tilde{\omega}, \tilde{M}]$ is the multinomial distr. associated to the $j$-th projected pseudo-center.
6: **end for**
7: **return** The cluster assignments $Cluster$.

---

still contain the full set of topics but the topic distribution will differ in the two: topics similar to the first pseudo-center will be predominant in the first subset, the others in the second. By recursively iterating this process, we obtain a binary tree where topic distributions in the nodes with higher depth are expected to be more concentrated on fewer topics.

In the next sections, we assess the validity of this approach on both synthetic and real-world data[6].

### 4.1 Experiment on Synthetic Data

In order to test the ability of SIDIWO to recover latent structures in data, we generate a dataset distributed as a single topic model (with a vocabulary of 100 words) whose 8 topics have an intrinsic hierarchical structure depicted in Figure 1a. In this figure, topics are on the $x$-axis, words on the $y$-axis, and green (resp. red) points represents high (resp low) probability. We see for example that the first 4 topics are concentrated over the 1st half of the vocabulary, and that topics 1 and 2 have high probability on the 1st and 3rd fourth of the words while for the other two it is on the 1st and 4th.

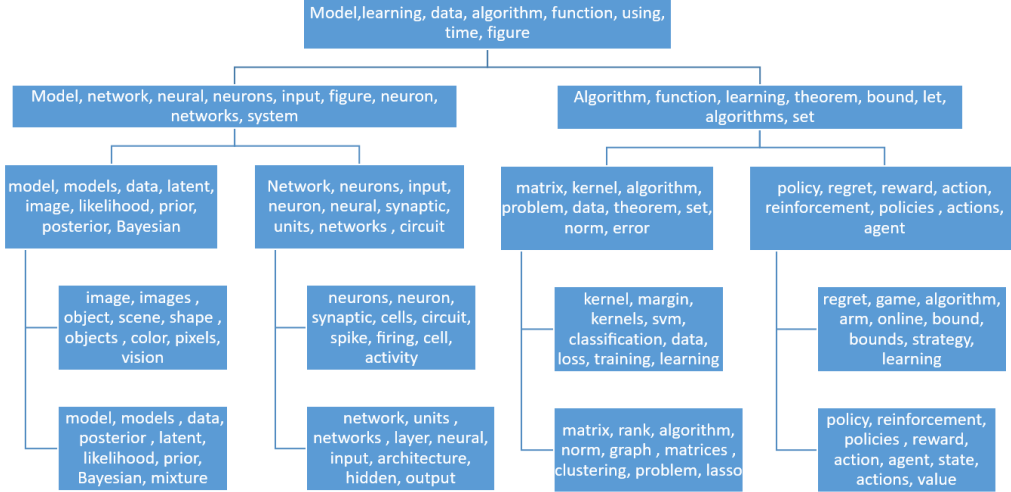

Figure 2: Experiment on the NIPS dataset.

We generate $400$ samples according to this model and we iteratively run Algorithm 2 to create a hierarchical binary tree with 8 leafs. We expect leafs to contain samples from a unique topic and internal nodes to gather similar topics. Results are displayed in Figure 1b where each chart represents a node of the tree (child nodes lay below their parent) and contains the heatmap of the samples clustered in that node ($x$-axis corresponds to samples and $y$-axis to words, red points are infrequent words and clear points frequent ones). The results are as expected: each leaf contains samples from one of the topics and internal nodes group similar topics together.

We compare the clustering accuracy of SIDIWO with other methods using the Adjusted Rand Index [27] of the partition of the data obtained at the leafs w.r.t the one obtained using the true topics; comparisons are with the flat clustering on $k = 8$ topics with TPM, the method from [3] (SVD), the one from [22] (Rand. Proj.) and ALS from [21], where ALS is applied to decompose a whitened $8 \times 8 \times 8$ tensor $T$, calculated as in Equation (3). We repeat the experiment 10 times with different random samples and we report the average results in Table 1c; SIDIWO always recovers the original topic almost perfectly, unlike competing methods. One intuition for this improvement is that each split in the divisive clustering helps remove noise in the moments.

## 4.2   Experiment on NIPS Conference Papers 1987-2015

We consider the full set of NIPS papers accepted between 1987 and 2015 , containing $n = 11,463$ papers [28]. We assume that the papers are distributed according to a single topic model, we keep the $d = 3000$ most frequent words as vocabulary and we iteratively run Algorithm 2 to create a binary tree of depth 4. The resulting tree is shown in Figure  2 where each node contains the most *relevant* words of the cluster, where the *relevance* [29] of a word $w \in \mathcal{C}_{node} \subset \mathcal{C}$ is defined by

$$ r(w, \mathcal{C}_{node}) = \lambda \log \mathbb{P}[w|\mathcal{C}_{node}] + (1 - \lambda) \log \frac{\mathbb{P}[w|\mathcal{C}_{node}]}{\mathbb{P}[w|\mathcal{C}]} \ , $$

where the weight parameter is set to $\lambda = 0.7$ and $\mathbb{P}[w|\mathcal{C}_{node}]$ (resp. $\mathbb{P}[w|\mathcal{C}]$) is the empirical frequency of $w$ in $\mathcal{C}_{node}$ (resp. in $\mathcal{C}$). The leafs clustering and the whole hierarchy have a neat interpretation. Looking at the leaves, we can easily hypothesize the dominant topics for the 8 clusters. From left to right we have: [image processing, probabilistic models], [neuroscience, neural networks], [kernel methods, algorithms], [online optimization, reinforcement learning]. Also, each node of the lower levels gathers meaningful keywords, confirming the ability of the method to hierarchically find meaningful topics. The running time for this experiment was 59 seconds.

## 4.3   Experiment on Wikipedia Mathematics Pages

We consider a subset of the full Wikipedia corpus, containing all articles ($n = 809$ texts) from the following math-related categories: linear algebra, ring theory, stochastic processes and optimization.

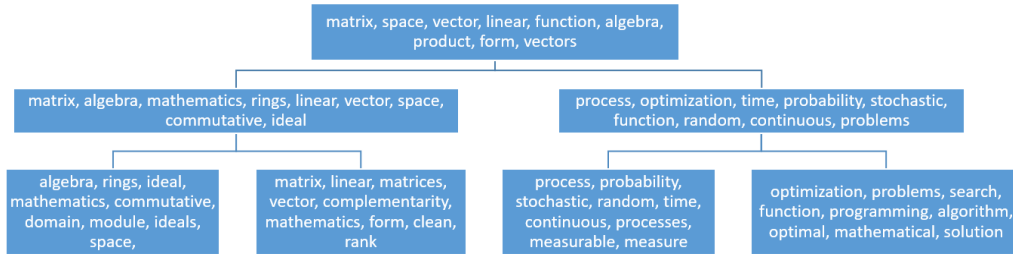

Figure 3: Experiment on the Wikipedia Mathematics Pages dataset.

We remove a set of 895 stop-words, keep a vocabulary of $d = 3000$ words and run SIDIWO to perform hierarchical topic modeling (using the same methodology as in the previous section). The resulting hierarchical clustering is shown in Figure 3 where we see that each leaf is characterized by one of the dominant topics: [ring theory, linear algebra], [stochastic processes, optimization] (from left to right). It is interesting to observe that the first level of the clustering has separated pure mathematical topics from applied ones. The running time for this experiment was 6 seconds.

# 5 Conclusions and future works

We proposed a novel spectral algorithm (SIDIWO) that generalizes recent method of moments algorithms relying on tensor decomposition. While previous algorithms lack robustness to model misspecification, SIDIWO provides meaningful results even in misspecified settings. Moreover, SIDIWO can be used to perform hierarchical method of moments estimation for latent variable models. In particular, we showed through hierarchical topic modeling experiments on synthetic and real data that SIDIWO provides meaningful results while being very computationally efficient.

A natural future work is to investigate the capability of the proposed hierarchical method to learn overcomplete latent variable models, a task that has received significant attention in recent literature [30, 31]. We are also interested in comparing the learning performance of SIDIWO the with those of other existing methods of moments in the realizable setting. On the applications side, we are interested in applying the methods developed in this paper to the healthcare analytics field, for instance to perform hierarchical clustering of patients using electronic healthcare records or more complex genetic data.

**Acknowledgments**

Guillaume Rabusseau acknowledges support of an IVADO postdoctoral fellowship. Borja Balle completed this work while at Lancaster University.

## Footnotes

*mruffini@cs.upc.edu

†guillaume.rabusseau@mail.mcgill.ca

‡pigem@amazon.co.uk

[4]See the supplementary material for an example corroborating this statement.

[5]The proofs of all the results are provided in the supplementary material.

[6]The experiments in this section have been performed in Python 2.7, using *numpy* [26] library for linear algebra operations, with the exception of the implementation of the method from [22], for which we used the author's Matlab implementation: https://github.com/kuleshov/tensor-factorization. All the experiments were run on a MacBook Pro with an Intel Core i5 processor. The implementation of the described algorithms can be found a this link: https://github.com/mruffini/Hierarchical-Methods-of-Moments.

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
