[Supplementary Material · hierarchical-methods-moments-full.pdf]

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

# A Low-rank whitenings may not admit a symmetric orthogonal decomposition.

In Section 2.2 we claimed that a symmetric tensor with CP-rank $k$, when whitened to a $l \times l \times l$ tensor $T_l$, may not admit a symmetric orthogonal decomposition if $l < k$. We give here a simple counter-example by constructing a tensor $M_3 \in \mathbb{R}^{3 \times 3 \times 3}$ whose $2 \times 2 \times 2$ whitening does not admit a symmetric orthogonal decomposition. We will make use of the following Lemma.

**Lemma A.1 ([32], Example 1.2.3)** *A $2 \times 2 \times 2$ symmetric tensor $T$ is orthogonally decomposable if and only if its entries satisfy the following equation:*

$$(T)_{1,1,1}(T)_{2,2,1} + (T)_{2,1,1}(T)_{2,2,2} = (T)^2_{2,1,1} + (T)^2_{2,2,1}. \tag{8}$$

Consider the following parameters[7]

$$\mu_1 = \begin{bmatrix} 1 \\ 0 \\ 0 \end{bmatrix}, \ \mu_2 = \begin{bmatrix} 0 \\ 1 \\ 0 \end{bmatrix}, \ \mu_3 = \begin{bmatrix} 1 \\ 1 \\ 1 \end{bmatrix}, \ \omega = \begin{bmatrix} 1 \\ 1 \\ 1 \end{bmatrix}$$

from which one can recover a matrix $M_2$ and a tensor $M_3$ from equations (1) and (2), both of rank $k = 3$. Using the top 2 singular vectors and values of $M_2$, $M_3$ would be whitened to a $2 \times 2 \times 2$ tensor $T$ with the following entries:

$$(T)_{1,1,1} = 2\left(\frac{1+\sqrt{3}}{2\sqrt{9+5\sqrt{3}}}\right)^3 + \left(\frac{2+\sqrt{3}}{\sqrt{9+5\sqrt{3}}}\right)^3,$$

$$(T)_{1,2,2} = (T)_{2,1,2} = (T)_{2,2,1} = \frac{1+\sqrt{3}}{2\sqrt{9+5\sqrt{3}}},$$

$$(T)_{1,2,1} = (T)_{1,1,2} = (T)_{2,2,2} = (T)_{2,1,1} = 0$$

One can check that in this case Eq. (8) is equivalent to $(T)_{1,1,1} = (T)_{2,2,1}$, which does not hold, hence $T$ is not orthogonally decomposable.

# B Proof Of Lemma 3.1

Consider the SVD of $M_2$:

$$M_2 = USU^\top$$

where $U \in \mathbb{R}^{d \times k}$ and $S \in \mathbb{R}^{k \times k}$ are obtained from the first $k$ singular vectors and values. Define now the matrix $E = US^{1/2}$; then there exists a unique $k \times k$ orthogonal matrix $O$ such that $M\Omega^{1/2} = EO$.

This implies that the slices of $M_3$ can be rewritten as follows:

$$M_{3,r} = M\,\Omega^{1/2} diag(m_r)\,\Omega^{1/2}\,M^\top = EOdiag(m_r)\,(EO)^\top.$$

Take now a generic matrix $D \in \mathcal{D}_k$; it can be written as

$$D = O_k^\top E^\dagger$$

for a certain orthonormal matrix $O_k$. So we have

$$DM_{3,r}D^\top = O_k^\top E^\dagger EOdiag(m_r)\,(EO)^\top (O_k^\top E^\dagger)^\top = O_k^\top Odiag(m_r)\,OO_k$$

This matrix is diagonal if and only if $O = O_k$, so the problem

$$\min_{D \in \mathcal{D}_k} \sum_{i \neq j} \left(\sum_{r=1}^{d} (DM_{3,r}D^\top)^2_{i,j}\right)^{1/2} \tag{9}$$

is optimized by $D = (M\Omega^{1/2})^\dagger = (EO)^\dagger$, which is the unique (up to a columns rescaling) feasible optimum.

# C    Proof Of Theorem 3.1

Let's recall the notation we are going to use. Consider the matrix $M_2$ and its SVD:

$$M_2 = USU^\top$$

For any $l \le k$, define $E_l = U_l S_l^{1/2} \in \mathbb{R}^{d \times l}$, where $U_l$ and $S_l$ are $U$ and $S$ truncated at the $l$-th singular vector (recall: $U \in \mathbb{R}^{d \times k}$ and $S \in \mathbb{R}^{k \times k}$). We know that there exists an orthonormal matrix $O$ such that

$$M \Omega^{1/2} = EO$$

Let's prove the various points of the Theorem.

1. Consider any matrix $D \in \mathcal{D}_l$. Then we will have, for an orthonormal $O_l$,

$$D = (E_l O_l)^\dagger = O_l^\top S_l^{-1/2} U_l^\top$$

   To prove the statement, it is enough to show that the matrix $C = DM\Omega^{1/2}$ has rank $l$. To see this, explicitly represent $C$:

$$C = DM\Omega^{1/2} = O_l^\top S_l^{-1/2} U_l^\top EO = O_l^\top S_l^{-1/2} U_l^\top U S^{1/2} O = O_l^\top \mathbb{I}_{l,k} O$$

   the fact that $O$ and $O_l$ are orthogonal proves the claim.

2. Consider again any matrix $D \in \mathcal{D}_l$, then

$$D^\dagger = \tilde{M}\tilde{\Omega}^{1/2} = (E_l O_l) = U_l S_l^{1/2} O_l$$

   The columns of $U_l$ are the left singular vectors of $M\Omega^{1/2}$, that span the best fit $l$-dimensional subspace of the space generated by the columns of $M$.

3. To prove this we will proceed by contradiction. Assume that $(M\Omega^{1/2} \mathbb{I}_{k,l})^\dagger \in \mathcal{D}_l$; this means that there exists an orthonormal matrix $O_l$ such that

$$M\Omega^{1/2} \mathbb{I}_{k,l} = E_l O_l = E\mathbb{I}_{k,l} O_l$$

   But $M\Omega^{1/2} = EO$, so

$$EO\mathbb{I}_{k,l} = E\mathbb{I}_{k,l} O_l$$

   This would imply that

$$\mathbb{I}_{l,k} O \mathbb{I}_{k,l} = O_l$$

   and so, for some $P \in \mathbb{R}^{k-l \times k-l}$

$$O = \left[ \begin{array}{c|c} O_l & 0 \\ \hline 0 & P \end{array} \right]$$

   Observe now that the matrix $Z = \Omega^{1/2} M^\top M \Omega^{1/2}$ has all the entries that are different from zero, by the hypothesis that $\langle \mu_i, \mu_j \rangle \ne 0$ for any $i, j$. However, we have that

$$Z = \Omega^{1/2} M^\top M \Omega^{1/2} = O^\top SO =$$

$$= \left[ \begin{array}{c|c} O_l^\top & 0 \\ \hline 0 & P^\top \end{array} \right] \left[ \begin{array}{c|c} S_l & 0 \\ \hline 0 & S_{l,k} \end{array} \right] \left[ \begin{array}{c|c} O_l & 0 \\ \hline 0 & P \end{array} \right] = \left[ \begin{array}{c|c} O_l^\top S_l O_l & 0 \\ \hline 0 & P^\top S_{l,k} P \end{array} \right]$$

   where $S_{l,k}$ is the diagonal matrix with the last $k - l$ singular values. So $Z$ has some zero entry. This contradiction proves the claim.

   The proof of the fact that $\mathbb{I}_{l,k}(M_\pi \Omega_\pi^{1/2})^\dagger \notin \mathcal{D}_l$ is identical.

# D   Proof Of Theorem 3.2

Recall the considered problem:

$$\min_{D \in \mathcal{D}_l} \sum_{i \neq j} \sup_{v \in \mathcal{V}_M} \sum_{h=1}^{k} \langle d_i, \mu_h \rangle \langle d_j, \mu_h \rangle \omega_h v_h \tag{10}$$

where

$$\mathcal{V}_M = \{v \in \mathbb{R}^k : v = \alpha^\top M, \text{where } \|\alpha\|_2 \leq 1\}$$

Consider any $v \in \mathcal{V}_M$, then it admits the following representation, for some $\alpha$ with $\|\alpha\|_2 \leq 1$:

$$v = [\langle \alpha, \mu_1 \rangle, ..., \langle \alpha, \mu_k \rangle]^\top.$$

This allows the following chain of equalities on the cost function:

$$\sum_{i \neq j} \sup_{v \in \mathcal{V}_M} \sum_{h=1}^{k} \langle d_i, \mu_h \rangle \langle d_j, \mu_h \rangle \omega_h v_h = \sum_{i \neq j} \sup_{\alpha : \|\alpha\|_2 \leq 1} \sum_{h=1}^{k} \langle \alpha, \mu_h \rangle \langle d_i, \mu_h \rangle \langle d_j, \mu_h \rangle \omega_h$$

$$= \sum_{i \neq j} \sup_{\alpha : \|\alpha\|_2 \leq 1} \sum_{r=1}^{d} \alpha_r \sum_{h=1}^{k} \mu_{h,r} \langle d_i, \mu_h \rangle \langle d_j, \mu_h \rangle \omega_h$$

$$= \sum_{i \neq j} \sup_{\alpha : \|\alpha\|_2 \leq 1} \sum_{r=1}^{d} \alpha_r (DM_{3,r} D^\top)_{i,j}$$

$$= \sum_{i \neq j} \sup_{\alpha : \|\alpha\|_2 \leq 1} \langle \alpha, t_{i,j} \rangle$$

$$= \sum_{i \neq j} \|t_{i,j}\|_2$$

Where the vector $t_{i,j}$ is defined as

$$t_{i,j} = ((DM_{3,1} D^\top)_{i,j}, ..., (DM_{3,d} D^\top)_{i,j})$$

and the last equality has been obtained from the fact that, for any vector $w \in \mathbb{R}^k$, we have

$$\|w\| = \sup_{\alpha : \|\alpha\|_2 \leq 1} \langle \alpha, w \rangle$$

This last equation proves our statement; in fact,

$$\sum_{i \neq j} \|t_{i,j}\|_2 = \sum_{i \neq j} (\sum_{r=1}^{d} (DM_{3,r} D^\top)_{i,j}^2)^{1/2}.$$

# E   Proof Of Theorem 3.3

First, observe that the set of $2 \times 2$ orthonormal matrices can be parametrized as

$$O_a = \begin{bmatrix} \sqrt{1-a^2} & a \\ -a & \sqrt{1-a^2} \end{bmatrix}, \quad \text{for } a \in [-1, 1]. \tag{11}$$

The set $\mathcal{D}_2$ can thus be rewritten in function of $a$, as

$$\mathcal{D}_2 = \{D : D = (E_2 O_a)^\dagger \text{ for } a \in [-1, 1]\}.$$

and Problem (5) can be rewritten as

$$\min_{a \in [-1,1]} \sum_{r=1}^{d} 2(O_a^\top H_{2,r} O_a)_{1,2}^2$$

where
$$H_{2,r} = E_2^\dagger M_{3,r} E_2^{\dagger\top}, \quad for \ r = \{1, ..., d\} \tag{12}$$

and where we used the fact that $O_a^\top H_{2,r} O_a$ is symmetric. We can then write

$$(O_a^\top H_{2,r} O_a)_{1,2}^2 = c_1^{(r)} a^4 + c_2^{(r)} a^3 \sqrt{1 - a^2} + c_3^{(r)} a \sqrt{1 - a^2} + c_4^{(r)} a^2 + c_5^{(r)}$$

where the coefficients can be written as

$$c_1^{(r)} = 4h^2 - f^2, \quad c_2^{(r)} = -4fh, \quad c_3^{(r)} = 2fh, \quad c_4^{(r)} = f^2 - 4h^2, \quad c_5^{(r)} = h^2$$

with $h = (H_{2,r})_{1,2}$ and $f = (H_{2,r})_{1,1} - (H_{2,r})_{2,2}$. Letting $c_j = \sum_{r=1}^d c_j^{(r)}$ for $j \in \{1, ..., 5\}$ it follows that optimizing Problem (5) is equivalent to minimize the following smooth real function

$$F(a) = c_1 a^4 + c_2 a^3 \sqrt{1 - a^2} + c_3 a \sqrt{1 - a^2} + c_4 a^2 + c_5.$$