[Reviews · NeurIPS 2017]

Reviewer 1



This paper studies the following problem: Can we make learning algorithms based on tensor decomposition robust to model misspecification? The authors propose to replace the tensor decomposition step by approximate simultaneous diagonalization. The problem studied here is clearly important. This paper does not really solve the problem, but makes reasonable progress, in particular at the experimental level for certain settings. I believe this is a reasonable paper for NIPS.

Reviewer 2



Spectral methods for learning latent variable models reduce the learning problem to the problem of performing some low-rank factorization over matrices collecting some moments of the target distribution. There are many variants of this in the literature, most approaches use SVD or other forms of tensor decompositions. This paper highlights an important limitation of many of these approaches. The main limitation is that most theoretical guarantees apply when the goal is to recover a model of k states assuming that the mixture distribution from which the moment matrix is computed also has k components. However, in many cases one is interested in learning models for l lower than k, what the authors call the misspecified setting. This paper proposes a method that performs joint diagonalization by combining a truncated SVD step with an optimisation. The theoretical analysis shows that this algorithm can work in the misspecified setting. The paper presents an application of the proposed algorithm to topic modelling. I believe this is a nice contribution, it is true that there are already many variants of the spectral method, each with its own claimed advantages. This been said, the one proposed in this paper is technically sound and addresses a true limitation of previous work. In terms of references the paper is quite complete, there are however a couple of relevant missing references: - B.Balle et al. "Local Loss Optimisation in operator models: A new insight into spectral learning"; since it is one of the first papers to frame spectral learning directly as an optimization. - More closely related to this submission, B. Balle et al "A Spectral Learning of Finite State Transducers" is to the best of my knowledge the first paper that proposes a spectral method based on joint diagonalization.